# Association between Residential Distance from Home to Hospital and Amputation of a Lower Extremity among Peripheral Artery Disease Patients in Japan

**DOI:** 10.3390/ijerph192013088

**Published:** 2022-10-12

**Authors:** Nobuko Hayashi, Yusuke Matsuyama, Takeo Fujiwara

**Affiliations:** Department of Global Health Promotion, Tokyo Medical and Dental University, Tokyo 113-8519, Japan

**Keywords:** peripheral artery disease (PAD), Chronic Limb Threatening Ischemia (CLTI), amputation, distance

## Abstract

Lack of access to care can lead to poor outcomes for patients with peripheral artery disease (PAD). We investigated the association between residential distance from home to hospital and amputation of the lower extremity among PAD patients in the Chiba peninsula, Japan. A retrospective cohort study with an average follow-up period of 2.96 years was conducted using data from 630 PAD patients who visited two hospitals in the Chiba peninsula, Japan, between 1 April 2010 to 31 March 2020. Information on disease status, residential address, and covariates was obtained from medical records. The association between amputation of a lower extremity, including toe amputation, and residential distance was evaluated by Cox proportional hazards model. Age, gender, Fontaine class, endovascular treatment, dialysis, diabetes mellitus, hypertension, dyslipidemia, current or past smoking, and aspirin use were adjusted. The median residential distance was 18.9 km (interquartile range, IQR: 22.1). Ninety-two patients (14.6%) underwent amputation of the lower extremity during the follow-up period. The longer residential distance was significantly associated with a higher risk of lower extremity amputation (hazard ratio per IQR = 1.35, 95% confidence interval, 1.01–1.82) after adjusting for covariates. Poorer access to a hospital, assessed as a longer residential distance from home to a hospital, was associated with amputation of the lower extremity among PAD patients.

## 1. Introduction

Peripheral artery disease (PAD) is a common cardiovascular disease [1]. The number of PAD patients increased from 164 million in 2000 to 202 million in 2010 globally due to the aging population and increased risk factors, such as diabetes mellitus (DM), dyslipidemia, renal dysfunction and smoking [2,3,4]. A previous systematic review reported that PAD is the third leading cause of atherosclerotic vascular morbidity after coronary heart disease and stroke [2]. Approximately 10–20% of PAD patients have intermittent claudication, another 50% have atypical leg symptoms, and those without exertional leg pain have poor mobility and a three-fold increase in mortality risk and major cardiovascular events [2,3].

One study in a rural area, Hokkaido prefecture, in Japan, reported that the prevalence of PAD was 1.6% for males and 0.7% for females for people under the age of 60 years, and 3.6% for males and 3.3% for females for aged 60 years or more [5]. Around 1–2% of PAD patients progress to critical limb ischemia (CLI) [6], which has poor prognoses with significant amputation rates as high as 40% [7]. Patients with poor prognoses require lower extremity amputation due to atherosclerotic PAD, whose acute mortality rate increases by 30% and a 5-year survival rate of less than 30% [8]. Some risk factors for lower extremity amputation among PAD patients, such as diabetes (DM), chronic kidney disease (CKD), ankle-brachial pressure index (ABI) < 0.4, cerebrovascular disease (CVD)/coronary artery disease (CAD) and history of lower extremity revascularization, have been reported [9,10].

While the distance from home to hospitals has been reported as a risk of poor postoperative outcomes in other surgical fields [11,12,13,14], only one study targeting CLI patients has been conducted in PAD patients in the US [15]. In the previous study, the data of 300 CLI patients who underwent revascularization between 2000 and 2010 at a single academic medical center in Utah was analyzed, and no significant difference was observed in amputation-free survival for CLI between those living more than 100 miles from a hospital and those living within 100 miles [15]. However, the findings may not be directly applicable to other countries because there is a variety of geographical accessibility and health insurance systems among countries. That is, the health insurance system may be a confounder of the association, as the geographical address is associated with socioeconomic status. Thus, testing the association between the residential address and the amputation of a lower extremity is essential among PAD patients in countries with universal healthcare coverage, such as Japan.

Southern Boso is a rural area, which is a part of the Chiba peninsula, Japan, with a population of 234,227 and an area of 1188 km^2^ (197 people/km^2^); 42% of the population is over 65 years old, which is higher than the national average of 28.4% in 2020 [16,17,18]. Only two hospitals provide treatment for PAD patients and lower extremity amputation surgery. Thus PAD patients in Southern Boso must travel a long distance to hospitals, usually driving a car alone. In such an area, the distance from home to the hospital would strongly influence access to care and outcomes of PAD. This study aimed to investigate the association between distance from home to hospital and lower extremity amputation outcomes among PAD patients in a rural area in Japan.

## 2. Materials and Methods

### 2.1. Study Participants

We retrospectively reviewed all charts for patients diagnosed with PAD between 1 April 2010, and 31 March 2020, at the following two hospitals, Awa regional medical center and Kameda medical center. A total of 904 patients were diagnosed with PAD. Patient demographics were obtained from medical records. After excluding 274 patients with missing information on demographic or clinical data, 630 patients were included in the analysis. This study was approved by Tokyo Medical and University Institutional Review Board. (M2020-007), Kameda Medical Center Institutional Review Board (20-004) and Awa Regional Medical Center Institutional Review Board (49).

### 2.2. Measurements

The outcome of this study was supported by data in terms of lower extremity amputation, including toe amputation, as obtained from medical records. The exposure variable, distance from home to hospital, was calculated by residential address using ArcGIS Pro version 2.7 (Esri, Redlands, CA, USA). Figure 1 shows the distribution of the residential distance from the hospital. Patients were stratified into two groups by median (18.9 km) to draw a Kaplan–Meier survival curve, while it was used as a continuous variable divided by IQR (22.1 km) in a Cox proportional hazard model. Other demographic and clinical variables comprised age, gender, pre-operational functional status (ambulatory, crutches/walker, wheelchair, bedbound), Fontaine class, endovascular treatment, bypass, complication, smoking status, family status, and medication were also obtained from the medical record.

### 2.3. Statistical Analysis

The Kaplan–Meier survival curve analysis and log-rank test were performed to evaluate the difference in amputation-free survival rate by the distance from home to the hospital divided by the median. The multivariable Cox proportional hazard model was conducted to investigate the association between the distance from home to the hospital and lower extremity amputation, including toe amputation. Risk factors of poor prognosis of PAD, including age, gender, Fontaine class, endovascular treatment, dialysis, diabetes mellitus, hypertension, dyslipidemia, current or past smoking, and aspirin use, were adjusted. STATA version 15 (StataCorp, College Station, TX, USA) was used for all analyses.

## 3. Results

Table 1 describes the demographic characteristics of the PAD patients compared with these groups (N = 630, 72.2% were male). There was no difference in amputation between groups (far: 57.6% vs. near: 42.4%, *p* = 0.114). Similarly, there were no significant differences in gender, preoperational functional status, DM, hypertension, dyslipidemia, coronary disease, atrial fibrillation, ulcer, family status, warfarin, cilostazol, Sarpogrelate hydrochloride, EPA, and Mecobalamin. There were significant differences in age, period, Fontaine class, endovascular treatment, bypass, dialysis, chronic renal failure (requiring less than 60 mL/min/1.73 m² in glomerular filtration rate), current or past smoking status, COPD, osteomyelitis, aspirin, statin, Clopidogrel, and Limaprost alfadex.

Figure 2 shows the lower extremity amputation-free survival curve. Patients whose residential distance from the hospital was 18.9 km or less had a higher amputation-free survival rate (*p* = 0.0537), although it was not statistically significant.

Table 2 shows the results from the Cox proportional hazards model. In a crude model, residential distance from the hospital was significantly associated with a higher risk of lower extremity amputation (hazard ratio, HR per IQR = 1.46; 95% CI, 1.08–1.98; *p* = 0.014). Fontaine class III (HR = 14.0; 95% CI, 1.77–111.0; *p* = 0.012), IV (HR = 173; 95% CI, 24.0–1244; *p* < 0.001), EVT (HR = 1.86; 95% CI, 1.20–2.89; *p* = 0.006), dialysis (HR = 1.84; 95% CI, 1.22–2.78; *p* = 0.004), diabetes mellitus (HR = 3.48; 95% CI, 2.16–5.62; *p* < 0.001), current or past smoking history (HR = 1.70; 95% CI, 1.035–2.80; *p* = 0.036) was significantly associated with higher risk of lower extremity amputation. In the multivariable model, residential distance from the hospital was significantly associated with a higher risk of lower extremity amputation (HR per IQR = 1.35; 95% CI, 1.01–1.82; *p* = 0.044). Fontaine class III (HR = 15.2; 95% CI, 1.88–123.4; *p* = 0.011) and IV (HR = 163; 95% CI, 22.1–1204; *p* < 0.001) were also significantly associated with lower extremity amputation.

## 4. Discussion

This study is the first to investigate the association between residential distance from the hospital and lower extremity amputation in PAD patients in countries other than the US and found that patients with longer residential distance from the hospital showed a higher risk of lower extremity amputation. 

A previous study in the US reported no association between residential distance from the hospital and lower extremity amputation in CLI patients [15]. The present study focused on PAD patients, including CLI patients. Different patient characteristics, health insurance systems, and geographic characteristics between US and Japan may explain the inconsistency between the present and previous studies. Previous studies in other surgical fields reported that increased travel distance from a patient’s home to the hospital was independently associated with an increase in hospital length of stay [11]. In another study on undergoing elective colorectal surgery, the increase in travel distance from a primary residence to the hospital was associated with an increase in length of stay [12]. Chou S. et al. found that patients living near an acceptable quality coronary artery bypass graft hospital traveled significantly less often and had lower in-hospital mortality rates if they were at higher risk [13].

Nemet GF et al. examined the association between distance from residence to hospital and health care utilization among elderly residents aged 65 or older in rural Vermont, US [19]. They reported that an increased distance from the physician’s office reduces utilization. Moreover, in a previous study, an increased travel burden was associated with a decreased likelihood of receiving adjuvant chemotherapy for colon cancer [20] and increased risk of anemia in chronic dialysis patients, especially the elderly, adjusted for socioeconomic status [21]. These results confirm that increased distance from a primary care physician reduced the utilization of hospital care. For elderly people, travel distance can be a barrier to visiting the hospital. Japan is a super-aged society [18], and there are restrictions on drivers licenses to prevent traffic accidents caused by elderly people [22,23,24]. Thus, even in Japan, where the whole population is covered by universal healthcare insurance, distance from residence to the hospital can be a barrier to access to care among older people.

In this study, most PAD patients had complications such as DM (49.1%), dialysis (31.4%), hypertension (70%), dyslipidemia (41.6%), chronic renal failure (46.7%), and coronary disease (44.8%). The complications associated with PAD progression and worsening lower extremity arterial perfusion are risk factors for PAD patients [4,25,26]. If a physician cannot regularly follow those patients, the complication will worsen [4]. For PAD patients with DM, preventative care and care coordination are critical in avoiding progression to Chronic Limb Threatening Ischemia (CLTI), ulceration, and eventually amputation [4]. Regularly, hemoglobin A1c testing, diabetic foot care, and vascular assessments, have been shown to help limit amputation in patients with diabetes and CLTI [27,28,29]. Providing home visit care should reduce the gap in rural areas with a high percentage of elderly people [30]. The findings also suggest that the physician should pay attention to PAD patients’ residential distance from the hospital.

This study has several limitations. First, the direct distance was used for every patient rather than road distance because the route to the hospital was unknown. Second, this study did not consider factors other than the residential distance that can influence access to care, such as access to transportation, socioeconomic status, available time for visiting the hospital, and support from family or friends. For example, the means of transportation to the hospital may influence the prognosis of PAD patients, but we were unable to consider it due to the lack of data. It would be necessary for future studies to consider these factors. Third, smoking was not associated with the risk of amputation in this study. Smoking is a risk factor for amputation among PAD patients [4]. We could not distinguish past and current smokers in the study participants because of a lack of information, which may have led to the different results on smoking status in the present study. Lastly, the findings’ generalizability is uncertain; this study’s results may not apply to patients in other healthcare regions, such as the urban area in Japan or other countries.

## 5. Conclusions

In Southern Boso, a rural area in Japan with a high proportion of elderly people and poor access to care, longer distances from home to the hospital were associated with lower extremity amputation among PAD patients. Other areas with a similar context include a high aging rate and limited healthcare resources. Access to care in rural areas must be improved for better outcomes for PAD patients living in rural areas.

## Figures and Tables

**Figure 1 ijerph-19-13088-f001:**
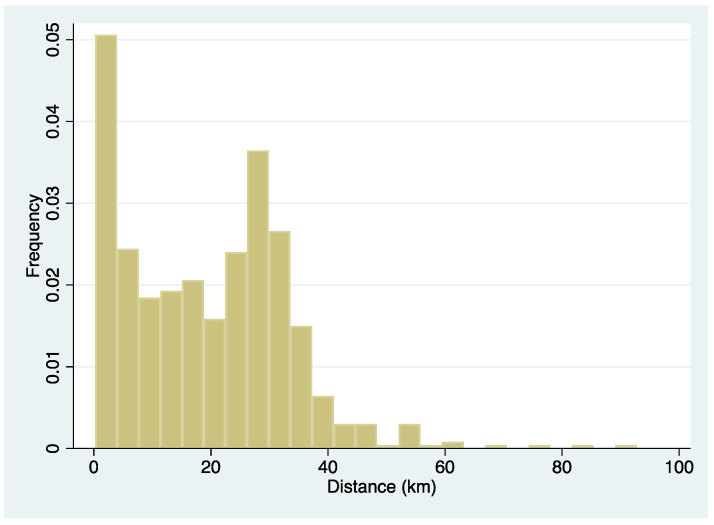
Histogram of the patient’s residential distance from the hospital.

**Figure 2 ijerph-19-13088-f002:**
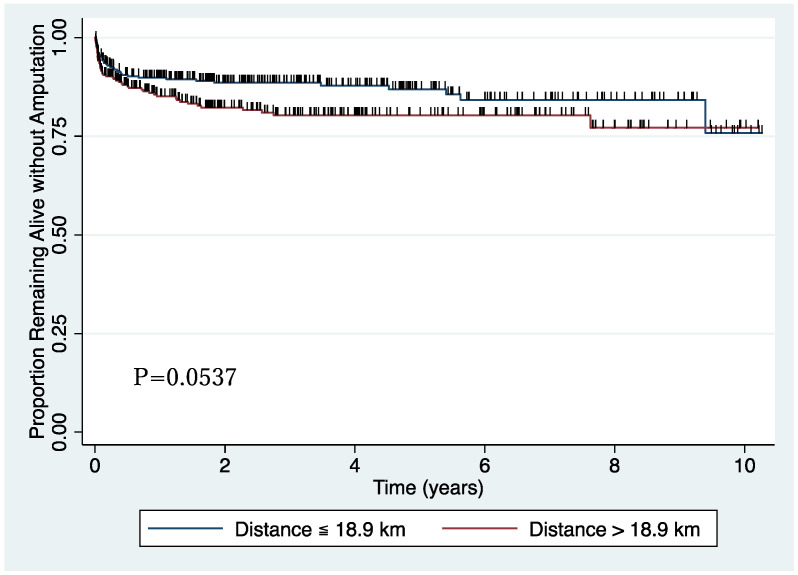
Amputation-free survival curve.

**Table 1 ijerph-19-13088-t001:** Patient characteristics (*n* = 630).

		Distance from Home to Hospital		
	All (*n* = 630)	≤18.9 km (*n* = 315)	>18.9 km (*n* = 315)	
	Mean (SD) or *n* (%)	Mean (SD) or *n* (%)	Mean (SD) or *n* (%)	*p* Value
Male gender	455 (72.2)	217 (47.7)	238 (52.3)	0.062
Age (years)	73.4 (10.9)	74.5 (11.3)	72.4 (10.4)	0.017
Period (year)	2.96 (2.71)	3.24 (2.72)	2.67 (2.68)	0.009
Amputation	92 (14.6)	39 (42.4)	53 (57.6)	0.114
Preoperational functional status				0.394
Ambulatory	485 (77)	240 (49.5)	245 (50.5)	
Crutches/walker	76 (12.1)	40 (52.6)	36 (47.4)	
Wheelchair	53 (8.4)	24 (45.3)	29 (54.7)	
Bedbound	16 (2.5)	11 (68.8)	5 (31.3)	
Fontaine class				<0.001
I	185 (29.4)	120 (64.9)	65 (35.1)	
II	180 (28.6)	63 (35)	117 (65)	
III	118 (18.7)	64 (54.2)	54 (45.8)	
IV	147 (23.3)	68 (46.3)	79 (53.7)	
Endovascular treatment	332 (52.7)	135 (40.7)	197 (59.3)	<0.001
Bypass	61 (9.7)	23 (37.7)	38 (62.3)	0.043
Dialysis	198 (31.4)	119 (60.1)	79 (39.9)	0.001
Diabetes mellitus	309 (49.1)	149 (48.2)	160 (51.8)	0.381
Hypertension	441 (70)	224 (50.8)	217 (49.2)	0.543
Dyslipidemia	262 (41.6)	131 (50)	131 (50)	1.000
Chronic renal failure	294 (46.7)	168 (57.1)	126 (42.9)	0.001
Coronary disease	282 (44.8)	133 (47.2)	149 (52.8)	0.200
Atrial fibrillation	133 (21.1)	73 (54.9)	60 (45.1)	0.204
Current or past smoking	432 (68.6)	195 (45.1)	237 (54.9)	<0.001
COPD	74 (11.8)	46 (62.2)	28 (37.8)	0.026
Ulcer	149 (23.7)	69 (46.3)	80 (53.7)	0.302
Osteomyelitis	86 (13.7)	34 (39.5)	52 (60.5)	0.037
Living with family	557 (88.4)	278 (49.9)	279 (50.1)	0.901
Aspirin	360 (57.1)	157 (43.6)	203 (56.4)	<0.001
Warfarin	106 (16.8)	54 (50.9)	52 (49.1)	0.831
Statin	298 (47.3)	133 (44.6)	165 (55.4)	0.011
Clopidogrel	178 (28.3)	69 (38.8)	109 (61.2)	<0.001
Cilostazol	179 (28.4)	79 (44.1)	100 (55.9)	0.064
Limaprost alfadex	61 (9.7)	43 (70.5)	18 (29.5)	0.001
Sarpogrelate hydrochloride	73 (11.6)	35 (48)	38 (52.1)	0.709
EPA	19 (3.02)	8 (42.1)	11 (57.9)	0.485
Mecobalamin	46 (7.31)	19 (41.3)	27 (58.7)	0.216

Data are represented as *n* (%) or mean (SD); Abbreviations: SD—standard deviation; COPD—chronic obstructive pulmonary disease; EPA—eicosapentaenoic acid.

**Table 2 ijerph-19-13088-t002:** Cox proportional hazards model for lower extremity amputation.

	Crude		Adjusted	
	HR (95% CI)	*p*-Value	HR (95% CI)	*p*-Value
Distance (per IQR)	1.46 (1.08–1.98)	0.014	1.35 (1.01–1.82)	0.044
Age	0.99 (0.97–1.004)	0.116	0.99 (0.97–1.01)	0.371
Male gender	1.60 (0.96–2.68)	0.073	1.13 (0.58–2.21)	0.717
Fontaine class				
I	Reference		Reference	
II	1.02 (0.064–16.3)	0.988	1.04 (0.63–17.1)	0.977
III	14.0 (1.77–111.0)	0.012	15.2 (1.88–123.4)	0.011
IV	173 (24.0–1244)	<0.001	163 (22.1–1204)	<0.001
EVT	1.86 (1.20–2.89)	0.006	0.94 (0.59–1.50)	0.806
Dialysis	1.84 (1.22–2.78)	0.004	1.12 (0.69–1.80)	0.652
Diabetes mellitus	3.48 (2.16–5.62)	<0.001	1.44 (0.84–2.45)	0.182
Hypertension	0.94 (0.60–1.46)	0.780	0.99 (0.63–1.55)	0.962
Dyslipidemia	0.86 (0.56–1.30)	0.466	0.99 (0.64–1.56)	0.994
Current or past smoking	1.70 (1.035–2.80)	0.036	1.18 (0.61–2.28)	0.622
Aspirin	1.45 (0.94–2.24)	0.092	0.78 (0.49–1.24)	0.286

Abbreviations: HR—Hazard ratio; CI—confidence interval; IQR—interquartile range = 22.1 km; EVT—Endovascular treatment.

## Data Availability

The data presented in this study are available on request from the corresponding author. The data are not publicly available due to privacy or ethical reasons.

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
