# Peer review of "Association between Residential Distance from Home to Hospital and Amputation of a Lower Extremity among Peripheral Artery Disease Patients in Japan"

_ijerph, 2022, doi:10.3390/ijerph192013088_

Round 1

Reviewer 1 Report

The authors are to be commended on the association between residential distance from home to hospital and amputation risk in PAD patients in Japanese rural area.

This article includes important results for future readers.

I have some comments below to be answered.

1. Page2, line64. In introduction, the authors mentioned important point about their study result. Patients in this study seemed to drive a car to hospital by themselves. This factor is important for PAD outcome because long time driving from home to hospital itself might be a risk for minor amputation in severe PAD patients. The way to the hospital in this study group should be discussed more in discussion.

2. Page3, line 124. Current or past smoking history have no relation on the risk of amputation in this study result. Smoking is reported as risk factor for lower limb amputation. The authors need to discuss about smoking in discussion session with some references. If the authors have some data about difference between current and past smoking in their patients, please show these data.

3. In discussion session. The reason or hypothesis about different results in amputation risk between this study and ref 14 is not discussed enough. How the authors think about reason of different result (different transportation, or insurance system)?  

Reviewer 2 Report

I wish to thank the Editor and authors for the opportunity to review this manuscript investigating access to care (as determined by distance) and amputation free survival in a population with peripheral artery disease. I was pleased to review this article as I believe that access to care is a significant barrier to health outcomes and this study has demonstrated as such in a rural part of Japan. The article was well written, succinct and clear. I have no concerns regarding methodology or analysis. The introduction and discussion sections were suitably concise given the analyses performed.

I have a couple of minor comments for the authors to consider, however, I would gladly recommend publication of this manuscript following minor revision.

Abstract:

Please state the following data (mean  2.94, SD 2.71) refers to years to amputation.

Introduction: 

Line 51: Delete, 'who'

Line 56: Should 'adders' be 'address'?

Line 57: can abbreviate PAD as already defined.

Discussion:

Line 152-154: Please revise. I think you intended to present findings from this study, however, the findings are not clearly presented.

Best wishes for your future research.

Round 2

Reviewer 1 Report

The authors revised the manuscript according to the reviewer’s comment.

Now, it is worth for publication in present form.